POINT OF VIEW

# COVID-19 as a catalyst for reimagining cervical cancer prevention

**Abstract** Cervical cancer has killed millions of women over the past decade. In 2019 the World Health Organization launched the Cervical Cancer Elimination Strategy, which included ambitious targets for vaccination, screening, and treatment. The COVID-19 pandemic disrupted progress on the strategy, but lessons learned during the pandemic – especially in vaccination, self-administered testing, and coordinated mobilization on a global scale – may help with efforts to achieve its targets. However, we must also learn from the failure of the COVID-19 response to include adequate representation of global voices. Efforts to eliminate cervical cancer will only succeed if those countries most affected are involved from the very start of planning. In this article we summarize innovations and highlight missed opportunities in the COVID response, and make recommendations to leverage the COVID experience to accelerate the elimination of cervical cancer globally.

REBECCA LUCKETT*, SARAH FELDMAN, YIN LING WOO, ANNA-BARBARA MOSCICKI, ANNA R GIULIANO, SILVIA DE SANJOSÉ, ANDREAS M KAUFMANN, SHUK ON ANNIE LEUNG, FRANCISCO GARCIA, KAREN CHAN, NEERJA BHATLA, MARGARET STANLEY, JULIA BROTHERTON, JOEL PALEFSKY, SUZANNE GARLAND, ON BEHALF OF THE INTERNATIONAL PAPILLOMAVIRUS SOCIETY (IPVS) POLICY COMMITTEE

**\*For correspondence:** rluckett@bidmc.harvard.edu

## Introduction

Cervical cancer is the fourth most common cancer in women (after breast, colorectal and lung cancer), and resulted in 311, 000 deaths in 2018 (*Arbyn et al., 2020*). The burden of cervical cancer also continues to rise – despite being a preventable disease – and this burden falls disproportionately on women in low- and middle-income countries (LMICs) (*Cohen et al., 2019*; *Zhang et al., 2021*). Many countries also lack the resources to treat cervical cancer, resulting in unnecessary death and suffering (*Union for International Cancer Control, 2023*). Moreover, cervical cancer is most often diagnosed in relatively young women who are often the primary wage earners in their household (*Zhang et al., 2021*; *American Society of Clinical Oncology, 2023*; *Reed et al., 2000*).

The goal of the Cervical Cancer Elimination Strategy, launched by the World Health Organization in 2019, is to reduce incidence of cervical cancer from about 15 per 100,000 women to less than 4 per 100,000 by the year 2030 (*World Health Organization, 2020*). The strategy has identified three targets to help it reach this goal: to provide HPV vaccination to 90% of girls by age 15; offer 70% women cervical screening with a high precision assay at least twice (by age 35, and again by age 45); and to treat 90% of women with pre-cancer and manage 90% of women with invasive cancer. HPV vaccination will likely have the biggest impact on the incidence of cervical cancer in the long run, with screening and treatment having bigger impacts in the short-term.

Unfortunately, progress on vaccination, screening and treatment was disrupted shortly after launch by the COVID-19 pandemic. In this article, on behalf of the Policy Committee of the International Papillomavirus Society (IPVS), we summarize the impact of the pandemic on efforts to eliminate cervical cancer and discuss how lessons learned during the pandemic can be applied to a different global public health threat – cervical cancer.

**Figure 1.** Map showing ever-in-lifetime cervical cancer screening coverage in women aged 30–49 years in 2019 by country. This map demonstrates that only 75 of the 202 countries surveyed in this study had achieved screening coverage of 70% or higher. Most countries in Africa and South Asia have lifetime screening coverage less than 20%. From *Bruni et al., 2022*. *Lancet Global Health* 10:e1115–1127. (CC BY 4.0).

## Cervical cancer prevention before the COVID-19 pandemic

Despite early demonstration of acceptability and feasibility, only 60% of WHO member states have incorporated HPV into their national vaccination schedule and only 13% of girls have completed HPV vaccination (*World Health Organization, 2022a*). Coverage had not reached the 90% target in most countries, attributed to various and different health systems capacity challenges (*Amponsah-Dacosta et al., 2020*). LMICs were also disadvantaged by limitations in the international supply chain, leaving many LMICs without adequate access to vaccine in the years leading up to the COVID-19 pandemic (*Garland et al., 2020*; *Colzani et al., 2021*).

The WHO elimination strategy ambitiously aims to increase global screening coverage to 70% with two HPV-based screens by the age of 45. Prior to the start of the pandemic, only 37% of countries had achieved lifetime screening coverage of 70% or higher, and none of these were low-income countries. Lifetime screening coverage was proportional to the country income strata with lifetime screening coverage reaching 70% in 58% of high-income countries, 28% of upper-middle-income countries, 6% of lower middle-income countries and 0% of low-income countries (*Figure 1*; *Bruni et al., 2022*). Within these numbers lie wide socioeconomic disparities within countries, particularly in opportunistic rather than population-based screening programs (*Phaswana-Mafuya and Peltzer, 2018*; *Corrêa et al., 2022*). Unsurprisingly, screening coverage is inversely correlated to incidence and mortality from cervical cancer (*Figure 2*; *Arbyn et al., 2018*).

## The impact of the COVID-19 pandemic on efforts to prevent cervical cancer

The pandemic impacted vaccination, screening and treatment of cervical cancer globally. In places with established HPV vaccination programs, there were severe interruptions, as school-based programs were hindered by school closures, routine clinical services were suspended for variable time frames and frequencies, and funds for HPV vaccination programming were redistributed (*Rao et al., 2022*; *Pillai, 2022*; *Daniels et al., 2021*). Additionally, roll-out of new HPV vaccination programs was delayed (*Newton, 2021*).

Similarly, there were drastic reductions in cervical cancer screening and treatment services across the globe related to the COVID-19 pandemic (*Mayo et al., 2021*; *Carcopino et al., 2022*; *Dennis et al., 2021*; *Istrate-Ofiteru et al., 2021*). Health facilities themselves had to limit care to sick patients and individuals feared the risk of exposure at health facilities (*Santos et al., 2021*; *Baaske et al., 2022*; *Schad et al., 2021*). COVID-19 resulted in greater disruptions in cervical cancer screening programs than some of the most damaging natural disasters (*Ortiz et al., 2021*). While some countries only experienced temporary disruption of services,

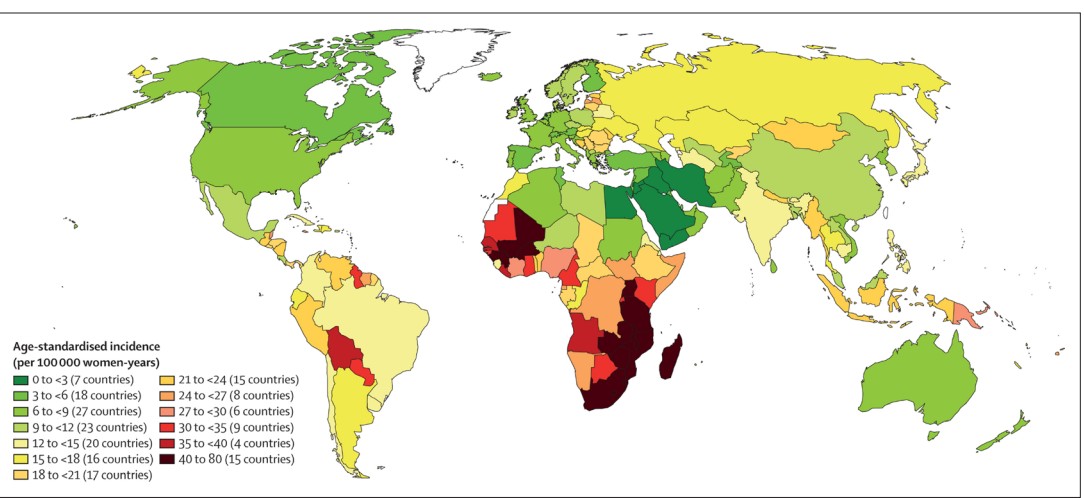

**Figure 2.** Map showing the age-standardised incidence of cervical cancer by country, estimated for 2018. The countries with the highest incidence of cervical cancer (in parts of sub-Saharan Africa, south Asia and South America) largely correspond to the countries with low screening coverage. From **Arbyn et al., 2020**. *Lancet Global Health* 8:e191–e203. (CC BY 4.0).

even after lifting COVID-19 related movement restrictions, screening remained lower in many places than baseline levels (**Basu et al., 2021**; **Liu et al., 2022**; **Miller et al., 2020**). There are little published data from LMICs, but it appears that preventive health services in LMICs took a severe hit and have been the slowest to resume baseline services (**Amouzou et al., 2021**; **Murewanhema, 2021**). Widespread burn-out has hindered efforts to return to pre-pandemic screening levels (**Smith and Perkins, 2022**).

We have yet to see how the disruption in health services due to the COVID-19 pandemic will impact cervical cancer burden on a global scale. Diagnoses of all cancers were reduced during the COVID-19 pandemic relative to pre-pandemic rates, resulting in a backlog of diagnoses (**Eskander et al., 2022**). A study in 2022 in Romania reported presentation of cervical cancer at more advanced stages during the pandemic, which accompanied significant changes in treatment courses due to interruptions in surgical and radiation services (**Popescu et al., 2022**). Recent modeling studies from the United Kingdom and United States estimate that the burden of cervical cancer will increase regardless of the length of time taken to catch up on missed screening, diagnosis and treatment due to COVID-19 (**Castanon et al., 2021b**; **Burger et al., 2021**). It is expected that greater disparity in these effects will be seen in settings without capacity to augment screening, diagnostic and treatment services to account for the backlog (**Bonadio et al., 2021**; **Castanon et al., 2021a**).

## Why was cervical screening so vulnerable during the pandemic?

Cervical cancer screening was particularly vulnerable to losing traction during the COVID-19 pandemic because at the start of the pandemic, screening in most countries required a pelvic examination. Pelvic examination is difficult to make "COVID-friendly". Counseling with examinations and treatment takes up to 30 minutes in a closed space. Disinfection protocols require additional time between patients. Waiting times for services are thus long, and a risk of exposure to a high volume of potential contacts is inherent to the process. Clinics were forced to compensate with reduced patient volume (**Sormani et al., 2021**).

In a review of cervical cancer screening guidelines in 139 countries across income strata, cytology was the most prevalent screening modality in HICs (78%), while visual inspection with acetic acid (VIA) was the most common method in LMICs and often a component of a screen-and-treat program (**Bruni et al., 2022**). While 35% (48 of 139) of country strategies noted plans to include HPV testing in their recommendations, only 17 countries had introduced HPV self-sampling into their national programs or guidelines by October 2020 and of those, half reserved the use of self-sampling for under-screened populations only. Countries with HPV self-sampling were mostly concentrated in HICs (**Figure 3**; **Serrano et al., 2022**).

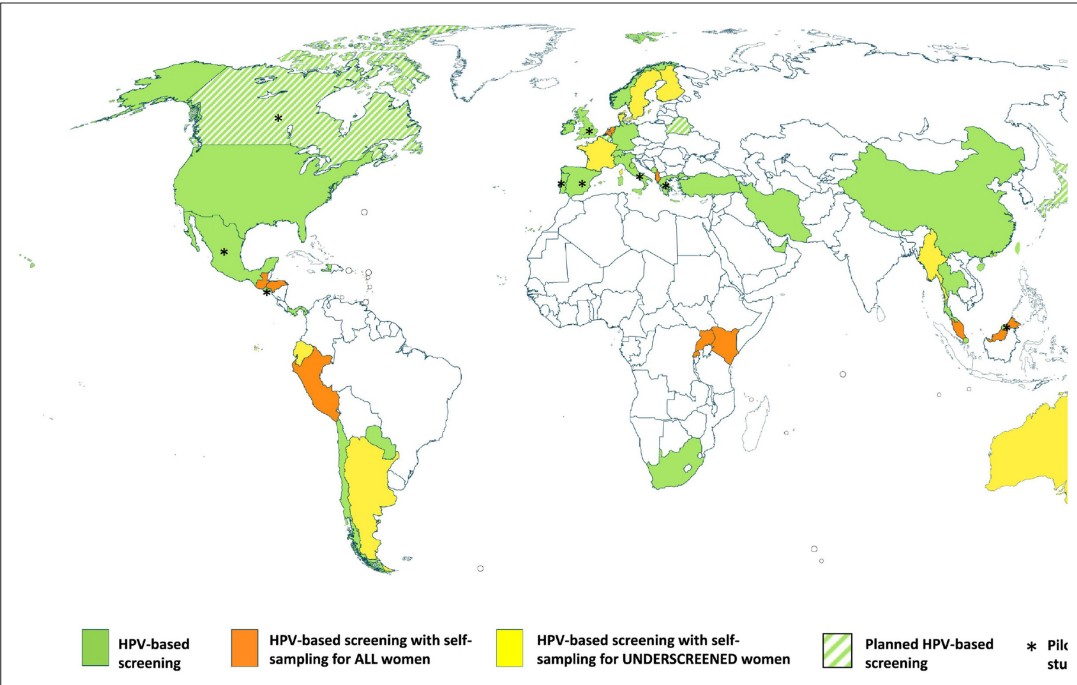

**Figure 3.** Map showing the self-sampling approach in countries that officially recommend HPV-based screening. As of October 2020, few countries had introduced HPV primary screening, and most were concentrated in high-income countries. Only 17 countries had introduced HPV self-sampling into their national programs or guidelines and of those, half reserved the use of self-sampling for under-screened populations only. From *Serrano et al., 2022*. *Preventive Medicine* 154:106900. (CC BY 4.0).

## The need for a new cervical cancer prevention paradigm

While it was clear prior to the COVID-19 pandemic that a new model for delivery of cervical cancer prevention services was needed, this need has been magnified post-pandemic by an unquantified and urgent backlog of screening and treatment services. Population-level cervical cancer screening with HPV self-testing is the most cost-effective approach in the long-run to addressing these issues, however the upfront affordability and the need for cost-effective triage strategies remain challenges (*Mezei et al., 2017*).

In considering cost-effectiveness, a critical consideration of outcomes is essential. Beyond the directs costs of implementation of cervical cancer programs lie the less tangible population-level economic, social and health costs of not improving screening. Programs focused on short-term measurable costs and outcomes likely miss the mark on establishing effective programming.

Perhaps the delay in focus on cervical cancer elimination due to the pandemic could be an opportunity to reframe cervical screening strategy with insight from the COVID-19 response. COVID-19 laid bare the existing inequities in global access to health resources,

from vaccination to treatment (*McGowan and Bambra, 2022*; *Pasquale et al., 2021*; *Stolberg, 2022*; *Foundation for Innovative New Diagnostics, 2023*). However, the COVID-19 joint resource mobilization effort provided a model for coordinated funding and policy development for vaccination, testing and treatment that was likely the most successful approach and may provide a template for a coordinated response to cervical screening (*World Health Organization, 2022b*).

Many countries failed to get enough COVID-19 vaccine in a timely fashion as high-income countries had most of the early global supply of vaccine *Tatar et al., 2021*. However, the acute need to rapidly scale up vaccination drove increased and more efficient manufacturing. Coordination of global partners allowed identification of bottlenecks in vaccine manufacturing and led to a variety of globally coordinated interventions, from creating international trade agreements, to procuring essential equipment, to developing and scaling-up manufacturing in countries without prior capacity (*World Health Organization, 2022c*).

In order to expand access to diagnostics, global efforts were made to diversify the supply chain and support emergency use listing of tests

to get them quickly to market. Global actors negotiated more affordable rapid diagnostics to increase global accessibility (*The Global Fund, 2022*). Countries rapidly implemented testing protocols, which were in many places available on demand without a health provider visit or referral. Screening outside of health facilities, often delivered in community settings and at home, and often done by individuals themselves became routine (*Feletto et al., 2020*). There was new engagement with the private sector for public health programming roll-out of both testing and vaccination.

There was a broad increase in individual self-efficacy in health engagement. The reporting of results directly to patients became routine. Population acceptance of swabs and self-testing was achieved through public health messaging, on-line and community-based communications (*Covid-19 Information Center, 2022*). There was increased utilization of telehealth, internet- and phone-based dissemination of accurate health information. Community health workers were engaged and programs were revitalized.

There was a movement to reduce the dependence of LMICs on global production by way of increasing testing capacity locally. This resulted in initiatives to start and scale up production of rapid diagnostics in Latin America, Africa and Asia, under their own branding and at more affordable prices (*UNITAID, 2021*). The great technical capacity of laboratories across the globe to engage in viral-based screening was front and center on the international stage. Twenty African countries were able to sequence SARS-CoV-2 by April 2020, and Botswana and South Africa were the first countries to detect the Omicron variant through genomic sequencing (*Viana et al., 2022*). Investment in training of both laboratory and bioinformatic personnel contributed to lasting and translatable capacity (*Tegally et al., 2022*).

## Recommendations for future action

In the rest of this article we will consider how the lessons learnt from the global response to the COVID-19 pandemic could be applied to efforts to eliminate cervical cancer globally, and we make 14 recommendations for further action (see *Box 1*).

The disruption of HPV vaccination programs will require a catch-up period with expanded age cohorts in order to reach children who missed vaccination during the global shortage and COVID-19 pandemic. These programs must be designed at national levels, according to national prespecified target ages groups for HPV vaccination. For countries that did not have established national HPV vaccination programming prior to the COVID-19 pandemic, logistical capacity created by COVID-19 vaccination programming may facilitate this process through having created stable supply chains and development of community-based vaccination teams (recommendation 1). Innovation and rapid development of COVID-19 vaccines may have created momentum to bolster the supply of effective prophylactic HPV vaccines. Increased supply and logistical capacity, coupled with new data supporting single dose HPV schedules may all facilitate the implementation of school-based and community-based HPV vaccination for both eligible children and those that require "catch up" vaccination (recommendation 2) (*The Lancet Oncology, 2022*; *Toh et al., 2021*). The development of COVID-19 vaccine registries may be capitalized upon to create vaccine registration for all age groups.

Learning from COVID-19 could accelerate implementation of primary HPV screening with self-testing both in clinical and community settings (*Arbyn et al., 2018*). There are currently multiple HPV assays on the market that can be used for self-sampling but await clinical validation (*Wentzensen et al., 2021*). The increase in production of molecular tests, the boom in molecular and nucleic acid testing platforms, alongside investment in laboratory capacity and expansion of manufacturing capacity for COVID-19 could contribute to increased capacity for HPV testing in the future (recommendation 3) (*Poljak et al., 2021*). Innovation in the development of point-of-care molecular diagnostics platforms designed to be used in non-laboratory setting by non-laboratory technicians may also facilitate broadly accessible test-and-treat cervical screening services (*Foundation for Innovative New Diagnostics, 2021*). Manufacturing diagnostics and vaccines to LMICs has the potential to expand accessibility of essential commodities for cervical cancer prevention (recommendation 4).

We have seen some examples of acceleration of HPV self-sampling programming in diverse settings, largely thanks to the disruption of routine services caused by COVID-19. Innovative strategies that had been successful in demonstration projects include community-based screening and mail-in HPV self-collection to maintain accessibility of cervical cancer screening services (*Woo*

## Box 1. Recommendations for accelerating the elimination of cervical cancer

1. Leverage supply chain opportunities developed during COVID-19 to distribute vaccines and diagnostics.
2. Maximize school-based and community-based vaccination for both eligible children and those that require "catch up" vaccination.
3. Invest in laboratory infrastructure and training to support HPV-based screening.
4. Establish local vaccine and diagnostic manufacturing.
5. Facilitate rapid evaluation and approval of HPV diagnostics.
6. Invest in diagnostic research and development to develop accurate point-of-care HPV tests.
7. Negotiate prices of HPV tests currently on the market to increase accessibility.
8. Increase investment on high-performance triage of positive HPV results.
9. Leverage social media, community networks and other indigenous methods for educating women and men about vaccination, screening and follow-up.
10. Use telehealth and web-based apps to report results and engage in follow-up.
11. Implement web-based electronic surveillance systems for cervical cancer screening program monitoring.
12. Evaluate programs on process indicators with the opportunity to rapidly respond to feedback and improve care.
13. Develop meaningful long-term outcomes for cervical cancer prevention, including health, economic and social metrics.
14. Support countries to develop tailored approaches to HPV-based screening and management.

*et al., 2022*; *Castle et al., 2011*; *Ejegod et al., 2022*; *Winer et al., 2019*; *Ngu et al., 2022*).

This momentum in HPV testing could move screening beyond a resourced-oriented approach and to high-performance HPV testing, or even test-and-treat, for all. As demonstrated by the COVID-19 diagnostic response, this will require a huge amount of coordinated work in research and development, product development, intellectual property management, pre-qualification processes, regional manufacturing, and in-country regulatory approval (recommendation 5) (*World Health Organization, 2022c*). Targeted investment in point-of-care HPV testing platforms and regulatory approval of self-sampling are needed to enable screen-and-treat programs to move to test-and-treat programs (recommendation 6) (*Lim, 2021*; *World Health Organization, 2022d*). Point-of-care HPV diagnostics would ideally replace visual inspection screening programs currently operating, and enable countries without existing cervical screening programs to launch screening services offering high-performance testing. Test-and-treat programs which offer same-day treatment of pre-cancerous lesions would be more efficient by narrowing the pool of women screened with HPV who then need pelvic examination for triage. Same-day treatment with either ablative or excisional procedures should concurrently be expanded, leveraging on-line or hybrid trainings and using of emerging technology such as artificial intelligence aided colposcopy. Coordinated innovation needs to be accompanied by price negotiations of currently available and pipeline diagnostics and treatment devices to ensure affordable testing and promote equitable distribution (recommendation 7).

Academic, program implementation and industry players need to coordinate efforts to optimize available triage strategies while also pushing the envelope in research and accessibility of novel triage strategies. Coordinated research on HPV triage with pooled genotyping, biomarkers, methylation, and artificial intelligence is necessary for effective programming (recommendation 8). Cost analysis at the country level using existing open-access tools will be necessary to ensure integration of more effective technologies (*Herrick et al., 2022*). Patient navigation to ensure that women who screen HPV positive return for triage and treatment is essential. Development of a risk-based approach to triage and diagnosis may further increase the efficiency and ability to implement effective screening programs (*Ribeiro et al., 2021*).

Cervical screening programs must capitalize on the momentum of individual engagement in attaining health services and health information that COVID-19 has created. Approaches to sustainably and actively engage women in cervical cancer prevention services and enhance their sense of autonomy in the process are essential. Information, education and communication needs to be innovative and tailored to how different groups obtain information across various settings. Information must be understandable directly by individuals without requiring complex explanations and delivered outside of the healthcare setting through community health workers, word-of-mouth, informal peer counseling and on-line platforms (*Eala and Tantengco, 2022*; *Ciceron et al., 2022*; *De Bocanegra et al., 2022*; *Christie-de Jong et al., 2022*). Working with traditional leaders and within established societal structures is critical to disseminating accurate information to communities and engendering trust in the screening process (recommendation 9).

Finally, COVID-19 forced the rapid development of web-based systems that could be adapted to engage individuals and monitor cervical screening programs (*Dixon et al., 2022*). These advances could create options that complement existing technological solutions that already exist. Brazil has demonstrated the success of an automated call and recall system to invite women due for screening, provide results directly and schedule follow-up appointments (*Corrêa et al., 2022*). Bangladesh monitored cervical screening during the pandemic through customization of the District Health Information Software (DHIS2), a platform that demonstrated its flexibility in adapting to pandemic tracking during the pandemic (*Basu et al., 2021*). Such technological innovation can optimize the implementation of population-level screening with HPV self-sampling and facilitate monitoring and evaluation of cervical cancer programs (recommendations 10 and 11) (*Woo et al., 2021*).

### Reimagining cervical cancer prevention

The success of cervical cancer elimination lies beyond innovation, technology and commodities – it must be one component of strong health systems with long-term strategies for managing endemic diseases. Integration into existing health systems was an area where the global coordinated response to COVID failed – it gathered international organizations without representation of global voices and failed to include sufficient perspective on delivery within variable health systems (*World Health Organization, 2022b*). A robust international network can support an equitable allocation of commodities and implementation for cervical cancer elimination. However, top-down global policy that is not contextualized in individual national planning is not effective long-term and perpetuates disparities. We need to examine the model by which externally-funded programs are designed and implemented, and focus on sustained success beyond project and strategy cycles. We need to work with countries to develop meaningful process indicators of successful programming that will lead to long-term progress towards eliminating cervical cancer. Pre-specified indicators must be subject to routine and rapid adjustment based on feedback that will improve programming (recommendation 12). Furthermore, long-term outcomes assessing progress on cervical cancer prevention, must include not only health indicators, but also consider economic and social metrics (recommendation 13).

Without meaningful country involvement, and national planning rooted in dynamic health systems, cervical cancer elimination will fail. Prioritizing national autonomy in designing programming that aligns with national health strategic goals, and then leveraging global financial, technical, and operational resources to support such programming is much more likely to be successful (recommendation 14). However, this global coordination will only succeed if those countries most impacted by cervical cancer are at the table.

## Conclusions

The COVID-19 pandemic demonstrated that it is possible to control an infectious disease when local, national, and global entities respond to the needs of the people they serve. The successes of the COVID-19 response have provided momentum and importantly, the failures have forced a re-evaluation of what meaningful engagement is. In this moment, we can reimagine equitable cervical cancer prevention for all and move forward on a more successful path towards elimination.

**Rebecca Luckett** is at the Beth Israel Deaconess Medical Center, Harvard Medical School, Boston, United States

rluckett@bidmc.harvard.edu

http://orcid.org/0000-0002-1975-8837

**Sarah Feldman** is at the Brigham and Women's Hospital, Harvard Medical School, Boston, United States

http://orcid.org/0000-0002-5582-9401

**Yin Ling Woo** is at the University of Malaya, Kuala Lumpur, Malaysia

http://orcid.org/0000-0003-1742-1066

**Anna-Barbara Moscicki** is at the University of California, Los Angeles, Los Angeles, United States

**Anna R Giuliano** is at the H Lee Moffitt Cancer Center and Research Institute, Tampa, United States

**Silvia de Sanjosé** is at the National Cancer Institute, Bethesda, United States and ISGlobal, Barcelona, Spain

http://orcid.org/0000-0002-5909-676X

**Andreas M Kaufmann** is at Charité - Universitätsmedizin Berlin Berlin, Freie Universität Berlin and Humboldt Universität zu Berlin, Berlin, Germany

http://orcid.org/0000-0001-7732-3009

**Shuk On Annie Leung** is at McGill University Health Center, Montreal, Canada

**Francisco Garcia** Deputy County Administrator for Community and Health Services and Chief Medical Officer, Pima County, Tuscon, United States

**Karen Chan** is at the University of Hong Kong, Hong Kong, China

**Neerja Bhatla** is at the All India Institute of Medical Sciences, New Delhi, India

**Margaret Stanley** is a Reviewing Editor at eLife and is at the University of Cambridge, Cambridge, United Kingdom

http://orcid.org/0000-0002-6865-6060

**Julia Brotherton** is at the Australian Centre for the Prevention of Cervical Cancer Melbourne, Australia

**Joel Palefsky** is at the University of California, San Francisco, San Francisco, United States

http://orcid.org/0000-0002-5097-3818

**Suzanne Garland** is at Melbourne Medical School, Royal Women's Hospital, Melbourne, Australia

*Author contributions:* Rebecca Luckett, Conceptualization, Writing – original draft; Sarah Feldman, Conceptualization, Writing – review and editing; Yin Ling Woo, Conceptualization, Writing – review and editing; Anna-Barbara Moscicki, Conceptualization, Writing – review and editing; Anna R Giuliano, Conceptualization, Writing – review and editing; Silvia de Sanjosé, Conceptualization, Writing – review and editing; Andreas M Kaufmann, Conceptualization, Writing – review and editing; Shuk On Annie Leung, Conceptualization, Writing – review and editing; Francisco Garcia, Conceptualization, Writing – review and editing; Karen Chan, Conceptualization, Writing – review and editing; Neerja Bhatla, Conceptualization, Writing – review and editing; Margaret Stanley, Writing – review and editing; Julia Brotherton, Writing – review and editing; Joel Palefsky, Writing – review and editing; Suzanne Garland, Conceptualization, Writing – review and editing

*Competing interests:* Rebecca Luckett: received a grant from NIH NCI 1K08CA271949-01. The author has no other competing interests to declare. Sarah Feldman: received grants from NCI/NIH and the Society to Improve Diagnosis in Medicine, and has received royalties from Uptodate. They have received payment for post-graduate talks at the Indian Health Service and Harvard Medical School, and for a community talk at Team Maureen. They received support for attending meetings of the ASCCP and the American Cancer Society. The author participated on the Mitre CDC sponsored initiative to integrate cervical cancer screening results into EHR and the American Cancer Society Advisory Committee ACS Cervical Cancer Roundtable. They are a Board Member of the IPVS and co-chair for ACS. The author has no other competing interests to declare. Yin Ling Woo: is a committee member for policy as part of IPVS. The author has no other competing interests to declare. Anna-Barbara Moscicki: has received from consulting fees from the Merck Advisory Board. The author participated on a Data Safety Monitoring Board/Advisory Board for CVIA 087 DSMB funded by PATH, and is an International Papillomavirus Society Board member. The author has no other competing interests to declare. Anna R Giuliano: has received grants and consulting fees from Merck & Co, Inc No other competing interests to declare. Silvia de Sanjosé: is a consultant at the National Cancer Institute (NIH, United States). No other competing interests to declare. Andreas M Kaufmann: has received grants from the EU EUROSTARS Program; has received payment for consultation from Paul-Ehrlich Gesellschaft e.V; has been issued with patent WO 2020/161285 A1 (Inventor); member of the Data Safety Monitoring Board/Advisory Board for the German Cancer Research Center (DKFZ). No other competing interests to declare. Karen Chan: member of Hong Kong SAR cancer coordinating committee (advisory board ) and the HK SAR cancer expert working group; President of the Hong Kong College of Obstetricians & Gynaecologists; council member of the Asian Society of Gynaecological Oncology (ASGO); board member for the Asia-Oceania Research Organisation in Genital Infection and Neoplasia. No other competing interests to declare. Margaret Stanley: Has received consulting fees from MSD Merck UK; has participated in a Global Advisory Board for HPV vaccines for Merck. No other competing interests to declare. Julia Brotherton: has received donated HPV tests and swabs for validation and research from Cepheid, Abbott, Seegene, Roche, AusDiagnostics, BD, and Copan. No other competing interests to declare. Joel Palefsky: has received grants from Merck & Co., Roche Diagnostics, Antiva Biosciences, Vir Biotechnologies and Virion Therapeutics; has received consulting fees from Merck & Co., Roche Diagnostics, Antiva Biosciences and Vir Biotechnologies; has received payment for consultation from Gilead Pharmaceuticals, Merck & Co. and Janssen Pharmaceuticals; has received

support for attending meetings from Merck & Co. and Roche Diagnostics; participates on the Data Safety Monitoring Board/Advisory Board for the IPVS; is Chair of the International HPV Awareness Day Campaign; has stock or stock options in Virion Therapeutics; has received resources/services from Atila Biosystems. No other competing interests to declare. Suzanne Garland: has received consulting fees and lecture fees from Merck; has participated in an advisory Board for Merck; is President of the International Papillomavirus Society; has received an education grant for a study of HPV in young women. No other competing interests to declare. The other authors declare that no competing interests exist.

### Funding

| Funder | Grant reference number | Author |
| --- | --- | --- |
| National Cancer Institute | K08CA271949 | Rebecca Luckett |

The funders had no role in study design, data collection and interpretation, or the decision to submit the work for publication.

**Decision letter and Author response**
Decision letter https://doi.org/10.7554/eLife.86266.sa1
Author response https://doi.org/10.7554/eLife.86266.sa2

## Additional files

### Supplementary files
• MDAR checklist

### Data availability
No data was generated.

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
