## [Decision Letter]

**Decision letter after peer review:**

Thank you for submitting your article "COVID-19 as a catalyst for reimagining cervical cancer prevention" to *eLife* for consideration as a Feature Article. Your article has been reviewed by two peer reviewers, and the evaluation has been overseen by an *eLife* Senior Editor (Eduardo Franco) and the *eLife* Features Editor (Peter Rodgers). The two reviewers have agreed to reveal their identity: Tim Palmer; Jesper Hansen Bonde.

The reviewers and editors have discussed the reviews and we have drafted this decision letter to help you prepare a revised submission. Please submit a revised version that addresses these concerns directly. Although we expect that you will address these comments in your response letter, we also need to see the corresponding revision clearly marked in the text of the manuscript. Some of the reviewers' comments may seem to be simple queries or challenges that do not prompt revisions to the text. Please keep in mind, however, that readers may have the same perspective as the reviewers. Therefore, it is essential that you amend or expand the text to clarify the narrative accordingly.

There are three points that need to be addressed.

1. The 13 recommendations you make are somewhat detached from the main text of the article. Please move the box containing the recommendations to earlier in the article and clearly state in the text that you are making 13 recommendations.

The article also needs to explain the reasoning behind each recommendation.

If the reasoning behind a given recommendation is already explained in the main text, please make this clear. One way to do this would be to add "(recommendation X)" to the end of the passage that explains the reasoning behind recommendation X.

And if the reasoning behind a recommendation is not explained in the main text, please add a few sentences to the text to explain the reasoning.

2. Please include some information/detail/opinion about the impact of COVID on the development and implementation of systems for 'point-of-care-testing-and-treatment'. (Please see the comment from reviewer#1 below for more on this). If you want to revise the article to address any of the other comments from reviewer#1, please do so.

3. The *eLife* Features Editor will also contact you separately about some editorial issues that you will need to address.

Summary:

This comprehensive and authoritative review examines the ways in which the experience of, and response to, COVID-19 could rejuvenate the drive for elimination of invasive cervical cancer as a public health problem worldwide. The authors argue compellingly that the technological and scientific response that enabled rapid development of vaccines and production of large volumes needs to be allied to organisational systems that enable delivery of both immunisation and screening solutions to the majority of countries that still have neither. They recognise that top-down systems need to be replaced by locally-relevant strategies for vaccine delivery and screening, so that they are effective and sustainable. Although the focus is on those countries with the greatest burden of disease, the proposals are relevant to high income countries too.

Comment from reviewer#1

The authors, who between them have many life-times of experience in this field, see the worldwide response to COVID-19 as a learning opportunity for those involved in cervical cancer elimination.

Having reviewed the stuttering progress to date towards the WHO target of eliminating cervical cancer, including the adverse effect of the COVID-19 pandemic on both vaccination and (established) screening programs, they point to the development of new vaccines and diagnostic tests for COVID-19 as examples of how global organisation can work well and, by extension, should be made to work well for HPV vaccines and screening tests. However, they also draw attention to the fact that COVID highlighted the problems of implementing cervical cancer elimination strategies in those countries that would benefit most. The problems were apparent before the pandemic, but are now in much sharper relief because of the pandemic-induced set-back. There is an opportunity to learn from the pandemic to improve the systems for delivery of vaccinations and screening at local level so that they are both more effective and resilient.

The problems that they are identify concern vaccine delivery and the provision of cost-effective screening. With regard to the former, they say that the cold-chain and communication strategies used for COVID are directly transferrable to HPV vaccination. Indeed, they may be easier to implement than COVID immunisation programmes as the target population is pre-adolescent and likely to be known to the education system.

The COVID communication infrastructure is more directly relevant to screening, as here a major problem is reaching women who need screening. Knowledge of local social and cultural factors is, as the authors indicate, vitally important. There are many examples from around the world, more than are cited by the authors, where sustainable engagement depends upon working with the grain of local cultures. Women may need 'permission' to attend for screening and then need peer-support in order to be tested. The verb 'be tested' is passive; women need to feel that 'being tested' is an active process; permission, peer-group support and, importantly, a testing process over which they have some control, are needed. This last criterion is also an area where COVID experience can be used to advantage.

Vaginal self-sampling has been proposed for almost a decade as being ideally suited to cervical cancer screening, given what is known about women's feelings about having cervical smears taken. It has taken this length of time for the evidence and technology to mature to a point where self-sampling, either using vaginal or urine specimens, has a solid evidence base. As the authors indicate, it has been adopted as a routine test, or a test for particular groups of women, in a few countries. Allied with molecular testing for hr-HPV, it offers a rapid and highly sensitive test of risk. However, the specificity of molecular tests, both for DNA and m-RNA, is too low and a triage method is needed to avoid unnecessary referrals for colposcopy. Cytology is not a feasible option in most settings as it requires a new sample and highly trained personnel to interpret the material.

At this point the authors point to the global, cross-discipline, cross-country approach that enabled the rapid development, assessment, licensure and deployment of technologies for rapid testing for COVID. Molecular triage strategies and point-of-care testing need to be developed, made available to and implemented in the great majority of countries that have no screening programmes. This is an area in which the high income countries have expertise and the necessary resources, used in the response to COVID and surely available for use of cervical cancer prevention.

'Point-of-care' is an important phrase. High income countries have the luxury of developed health-care systems which encompass most of the population; low and middle income countries are not so blessed. It is unethical to screen if there is no realistic prospect of treatment, and in most settings, treatment is divorced from the screening process. Cervical cancer elimination in those countries most affected will be contingent upon linking screening and treatment into one seamless process. Some point-of-care testing platforms are available, but there needs to be a greater variety. Small-scale projects are being run, for example in Malawi, where screening, assessment and treatment all take place at the same visit. Experiences from these projects needs to drive the research and development of suitable equipment that will deliver true point-of-care screening. Again, high-income countries have the resources of drive this forward.

Only then, in combination with harnessing the lessons of COVID with regard to vaccine delivery, to communication with individuals, to self-sampling, and to recording activity and outcomes, will the WHO target become a realistic one in the timescale envisaged.

---

## [Author Response]

There are three points that need to be addressed.1. The 13 recommendations you make are somewhat detached from the main text of the article. Please move the box containing the recommendations to earlier in the article and clearly state in the text that you are making 13 recommendations.The article also needs to explain the reasoning behind each recommendation.If the reasoning behind a given recommendation is already explained in the main text, please make this clear. One way to do this would be to add "(recommendation X)" to the end of the passage that explains the reasoning behind recommendation X.And if the reasoning behind a recommendation is not explained in the main text, please add a few sentences to the text to explain the reasoning.

This is really helpful feedback and also helped us to reorder the recommendations so that they flow with the body of the manuscript. We added point 8 which on reflection, we believe needs attention and should be included – we have thus increased our recommendations to 14.

2. Please include some information/detail/opinion about the impact of COVID on the development and implementation of systems for 'point-of-care-testing-and-treatment'. (Please see the comment from reviewer#1 below for more on this). If you want to revise the article to address any of the other comments from reviewer#1, please do so.

We have included further suggestions guiding test-and-treat policy on p13.

The feedback from reviewer 1 is really helpful. We have included more clear points related to school-based HPV vaccination and sustainable engagement and control of healthcare; as well as developed the test-and-treat concept more fully in the manuscript. We do want to highlight our sentiment that current diagnostics for HPV testing are not true point-of-care that will enable successful population-level test-and-treat. His example in Malawi is a site that uses the Xpert HPV device. While we know that in a research setting, many sites have been able to achieve same day test-and-treat of relatively small numbers of women with non-lab based Xpert HPV devices (usually 4 chamber devices), our experience in implementing national cervical cancer prevention programs (including relatively well-resourced LMIC settings like Botswana) is that the Xpert device does not meet full specifications for point-of-care diagnostics, and is not likely a solution to implementation of population-level cervical screening in most settings.

We are open to the editor’s thoughts if other points should be added/clarified based on this feedback. Also, our sense is that reviewer 1’s feedback could make for a good editorial on the full manuscript.